# Microstructural Changes and Mechanical Properties of Precipitation-Strengthened Medium-Entropy Fe_71.25_(CoCrMnNi)_23.75_Cu_3_Al_2_ Maraging Alloy

**DOI:** 10.3390/ma16093589

**Published:** 2023-05-07

**Authors:** Unhae Lee, Jae Wung Bae

**Affiliations:** 1POSCO Technical Research Laboratories, Gwangyang 57807, Republic of Korea; uhlee@bistep.re.kr; 2BISTEP Evaluation & Analysis of Regional Innovation Program Division, Busan 48058, Republic of Korea; 3Department of Metallurgical Engineering, Pukyong National University, Busan 48513, Republic of Korea

**Keywords:** high-entropy alloy, medium-entropy alloy, age-hardening, mechanical properties, B2 precipitates

## Abstract

Metal alloys with enhanced mechanical properties are in considerable demand in various industries. Thus, this study focused on the development of nanosized precipitates in Fe_71.25_(CoCrMnNi)_23.75_Cu_3_Al_2_ maraging medium-entropy alloy (MEA). The Fe-based alloying design in the MEA samples initially formed a body-centered cubic (BCC) lath martensite structure. After a subsequent annealing process at 450 °C for varying durations (1, 3, 5, and 7 h), nanosized precipitates (B2 intermetallic) enriched with Cu and with a diameter of approximately 5 nm formed, significantly increasing the hardness of the alloy. The highest Vickers microhardness of 597 HV, along with compressive yield strength and ultimate compressive strength of 2079 MPa and 2843 MPa, respectively, was achieved for the Aged_7h sample. Therefore, the BCC lath martensite structure with B2 intermetallics leads to remarkable mechanical properties.

## 1. Introduction

Recently, the demand for high-strength and high-fracture-toughness materials in the transportation industry, particularly in aerospace, automobile, and naval applications, has rapidly increased [1,2,3]. Maraging steel exhibits excellent mechanical properties owing to the combination of a martensite structure and precipitates [1,2,3,4,5,6,7].

Medium- and high-entropy alloys (MEAs and HEAs, respectively) are promising materials with superior mechanical properties. Among many MEA and HEA systems, alloys consisting of Co, Cr, Fe, Mn, and Ni are among the most widely studied HEAs [8,9,10,11]. The classification of MEA or HEA depends on the mixing entropy (Δ*S_mix_*) caused by multiple elements in an equiatomic (or near-equiatomic) fraction of 5–35 at.%; Δ*S_mix_* can be calculated using the following equation [9,12]:(1)$${\Delta S}_{mix}=-R\sum_{i=1}^{n}X_{i}lnX_{i}$$
where *R* is the gas constant. Generally, an alloy can be classified as the HEA when the Δ*S_mix_* value exceeds 1.61*R*. Alternatively, a Δ*S_mix_* deal between 1.61*R* and 0.69*R* would classify an alloy as an MEA [13]. MEA and HEA systems have excellent strength–ductility balance with high phase stability over a wide temperature range owing to their unique composition [11,14].

The phase stability of MEAs and HEAs is attributed to the complex interaction between the constituent elements, which decreases the kinetics at high temperatures [13]. Hence, MEA and HEA tend to form simple solid solution-type structures that are either face-centered cubic (FCC), body-centered cubic (BCC), or mixed [15,16]. Among these phases, BCC-structured alloys exhibit high hardness and compressive strength [17].

The present work focuses on implementing an aging treatment based on the concept of maraging steel in a non-equiatomic MEA system of Fe_71.25_(CoCrMnNi)_23.75_Cu_3_Al_2_ (at.%). This concept was initially proposed to enhance the mechanical properties of alloys. According to Equation (1), this type of alloy has a Δ*S_mix_* value of 1.09*R*; hence, it is categorized as an MEA.

Recently, researchers have extensively studied MEAs and HEAs due to the intriguing mechanical properties that arise from their unique compositional characteristics. However, these HEAs possess limitations, particularly in terms of their ductility and fracture toughness, which have motivated researchers to explore alternative alloying systems. In light of these limitations, our study focuses on a newly developed Fe-based medium-entropy alloy (MEA), Fe_71.25_(CoCrMnNi)_23.75_Cu_3_Al_2_, designed to offer a superior combination of strength, ductility, and toughness. The selection of this alloying system was based on the following considerations: (1) the Fe-based alloying design aimed to achieve a body-centered cubic (BCC) lath martensite structure, which is known for its superior mechanical properties [18]; (2) the addition of Cu and Al was intended to promote the formation of nanosized B2 intermetallic and Cu-rich precipitates during annealing, which could further improve the hardness, strength and toughness of the alloy [19,20,21]; and (3) the Fe based CoCrMnNi medium-entropy alloy was incorporated to enhance the overall performance of the material due to its reported high strength and ductility [22]. Therefore, this study was designed to understand the microstructural changes and mechanical properties of precipitation-strengthened medium-entropy Fe_71.25_(CoCrMnNi)_23.75_Cu_3_Al_2_ maraging alloy during the annealing process.

## 2. Materials and Methods

Fe_71.25_(CoCrMnNi)_23.75_Cu_3_Al_2_ (at.%) alloy was fabricated in a vacuum arc furnace under an inert Ar gas atmosphere. The as-cast ingot was homogenized at 1000 °C for 24 h in an Ar + H_2_ (3%) atmosphere, followed by cold rolling at room temperature to achieve a thickness reduction of up to 90%, thus eliminating the as-cast structure. The specimen was then recrystallized at 900 °C for 2 h to induce recrystallization and eliminate the effects of casting defects [22]. This was done to obtain a uniform microstructure before water quenching to room temperature to promote the formation of the martensite phase (Aged_0h sample). Finally, the annealing process was conducted at 450 °C for 1, 3, 5, and 7 h (labeled as Aged_1h, Aged_3h, Aged_5h, and Aged_7h, respectively) to provide a comprehensive analysis of the microstructure and mechanical properties evolution with different aging times. To investigate the evolution of hardness values with respect to annealing time, we conducted more than ten micro-Vickers hardness measurements on each sample using a 1 kgF load on the transverse-direction plane of the rolled plate after the annealing process. The compression test was performed at room temperature using a cylindrical specimen of diameter 4 mm and length 6 mm. The initial strain rate for the compression test was 8.3 × 10^−4^ s^−1^. Phase analysis was performed by applying X-ray diffraction (XRD) with observations ranging from 40° to 100° at a step size of 0.026°. Electron backscatter diffraction (EBSD) was used to observe microstructure development. Transmission electron microscopy (TEM) and atom probe tomography (APT) were used to analyze the martensite structure and precipitate formation.

## 3. Results and Discussion

### 3.1. Mechanical Property Analysis

The Vickers hardness of the Fe_71.25_(CoCrMnNi)_23.75_Cu_3_Al_2_ (at.%) MEA during the annealing process is shown in Figure 1. Initially, the hardness value of the Aged_0h specimen was measured to be approximately 328 HV. However, after annealing, the hardness value increased significantly as the annealing time increased from 1 to 7 h. The highest hardness value was 597 HV, which was observed after 7 h of annealing; this value was nearly twice that of the Aged_0h specimen. This significant increase in hardness may indicate the formation of precipitation hardening after annealing. According to Tewari et al. [23], the annealing process tends toward the growth of precipitate particles and intensifying their spatial correlations. Therefore, it causes the hardness to increase as the aging process is prolonged.

A compression test was performed on the Aged_7h specimen to evaluate its mechanical properties further. As shown in Figure 2a, the specimen exhibited a brittle fracture phenomenon after compression, related to a high Vickers hardness value. As reported by Pesicka et al. [24], the brittle phenomenon is often correlated with the existence of a BCC phase; thus, it may indicate the presence of a BCC phase in the specimen. In this study, the engineering stress–strain curve result revealed that a high compressive yield strength (σ_y_) of 2079 MPa, high ultimate compression strength (σ_UCS_) of 2843 MPa, and elongation (ε) of 18% were successfully obtained. The high values of σ_y_ and σ_UCS_ are interesting because the values obtained were higher than those obtained in previous studies on other typical maraging steels [25]. Furthermore, in Figure 2b, the studied alloy also exhibited the remarkable advantage of having a lower raw material cost than other commercial maraging steels [5,23,26]. The price calculation of the raw material cost (USD/kg) of the metal entered in Figure 2b was based on the price of pure metal with a purity of 99.9% or higher based on the content of the element contained in the London metal exchange. The London metal exchange shows the price trend of the most reliable materials internationally.

### 3.2. Phase Analysis by XRD

The XRD patterns in Figure 3 display the phase analysis of the Aged_0h and Aged_7h samples. After the recrystallization process, the alloy consisted of a single BCC structure, whereas, after aging, the XRD patterns revealed BCC and FCC structures. The BCC structure remained stable even after recrystallization and aging because of the presence of Fe and Al as BCC stabilizers [27,28,29]. Moreover, the additional peaks of the FCC phase imply the formation of a precipitate after aging. The Cu content of the alloy promoted this precipitate formation. A similar precipitate formation was also presented in a previous report on Fe-based alloys by Wen et al. [30], which demonstrated that the Cu content controlled the precipitate nucleation and caused the ε-copper precipitate to possess an equilibrium FCC structure. Therefore, the combination of the BCC phase and FCC precipitate in the Aged_7h sample may have increased the hardness value (Figure 1) and the compression strength (Figure 2a). Further investigation of the microstructure was performed using EBSD.

### 3.3. Microstructure and Precipitation Analysis

#### 3.3.1. Microstructure Analysis by EBSD

To confirm the microstructure after recrystallization and aging, EBSD analysis was performed; the results are shown in Figure 4a–f. The image quality (IQ) maps of the Aged_0h and Aged_7h samples in Figure 4a,d show groups of lath structures within the grain, which indicate the formation of lath martensite. According to the phase maps in Figure 4b,e, both samples were dominated by the BCC phase, and some areas exhibited an FCC structure. This result is in good agreement with the XRD results shown in Figure 3. The BCC phase was mainly observed within the grains. Therefore, the matrix of the alloy consisted of a BCC structure. The inverse pole figure (IPF) maps of the Aged_0h and Aged_7h samples in Figure 4c,f show groups of lath martensite that exhibited the same plane within the grain (martensite packet). Each martensite packet consisted of several blocks (a group of laths with the same orientation). A random distribution of the block structure, indicated by the color variation, was observed within the grains of the Aged_0h sample. Alternatively, the martensite packets and blocks in the Aged_7h sample exhibited a nearly uniform contrast in the (111) plane. It suggests that the martensite blocks and packets in the Aged_7h sample were thicker than those in the Aged_0h sample. These thick martensite blocks and packets may facilitate plastic strain accommodation during deformation and help to improve the strength of the alloy [31]. The Cu-rich FCC structure displayed in Figure 3 is not clearly seen in Figure 4e since the size of Cu-rich FCC precipitation with a diameter of approximately 5 nm is much finer than the step size of EBSD measurement.

#### 3.3.2. Martensite Microstructure Analysis by TEM

Figure 5a shows a TEM bright-field (BF) image, illustrating many lath martensite grains in the Aged_7h sample. The lath martensite structures were confirmed by TEM as a BCC phase; the corresponding selected area diffraction patterns (SADPs) are shown in Figure 5b,c. The zone axes of the SADPs in Figure 5b,c were along ferrite [$\bar{1}01$] and [001], respectively. The SADP indicates the existence of twins in the ferritic lath martensite and the appearance of extra diffraction spots from double diffraction (Figure 5b). The SADP shown in Figure 5b was obtained from the twin [$0\bar{1}1$] and ferrite [$\bar{1}01$] directions. The diffraction spots connected by dashed white lines correspond to one ferrite grain (twin, t), whereas those connected by solid white lines correspond to the other ferrite grains (α). The twinning relationship between the ferrite twin (t) and ferrite grain (α) was [$0\bar{1}1$]_t_//[$\bar{1}01$]_α_ and (211)_t_//(121)_α_. The Miller indices of one ferrite grain in Figure 5b have been underlined to distinguish them from those of other ferrite grains. The representative twin planes of the BCC crystal were found to be the (211) planes of the ferrite twin and the (121) planes of the ferrite. According to Yu et al. [1], the combination of twinning and lath martensite structures in maraging steel results in superior strain-hardening capacity and high ductility.

Figure 5c shows the SADP of ferrite lath martensite. The zone axis of the ferrite in the SADP was along the [001] direction. Therefore, the major reflections with underlines corresponded to ferrite. Initially, TEM in situ observation could not reveal any extra reflections in the SADP because the reflections from the ferrite laths were excessively strong. However, after careful examination of the [001] SADP, other reflections were discovered: four extra reflections located near the (100) position. These extra reflections with very weak intensities were barely observable. A dark-field (DF) image taken from the (100) position was displayed to analyze the probability of nanoprecipitates in the Aged_7h sample.

#### 3.3.3. Precipitate Formation Analysis by TEM and APT

The DF TEM image in Figure 5d, as obtained for the extra reflection near the (100) position shown in Figure 5c, shows the uniform distribution of fine coherent nanoprecipitates in the ferrite lath martensite. The DF image was captured after a relatively long TEM exposure time because the (100) reflection was excessively weak. The extra reflections observed in the TEM observations generated fine coherent nanoprecipitates in the ferrite lath martensite. This phenomenon could be attributed to the clustering of the Cu atoms acting as nucleation sites, which accelerated the formation of precipitate particles [22,30]. In a previous study, Stiller et al. [32] reported that the precipitation in 9Ni-12Cr-2Cu-4Mo (wt.%) maraging steel begins with the formation of Cu-rich particles, which act as nucleation sites for a Ni-rich phase of Ni_3_(Ti, Al). Moreover, Miller et al. [33] found that Mn could be detected in Cu-enriched precipitates in a specific type of Fe-Cu-Mn alloy. Thus, an ultrahigh ultimate compressive strength can be achieved for Cu precipitation-strengthened steels by properly adding Mn, Ni, and Al. According to these reports, the precipitation shown in Figure 5d can be assumed to be Cu-Ni-Mn-Al co-precipitates [30]. Previously, this nanoprecipitate structure could not be observed in XRD patterns (Figure 3); this may be attributable to the small fraction of precipitate and the peaks of the B2 and BCC phases are very similar [4,22,26,34].

Further observation of the precipitate was conducted using APT analysis, with Figure 6a–h showing the three-dimensional (3D) APT results with ion maps for each element. Heterogeneity was observed in the Cu ion map, as shown in Figure 6h. Unlike the other ion maps, the Cu ion map had a nearly spherical morphology. This phenomenon further indicates the tendency of Cu to form clusters [34]. To delineate the precipitate origin, quantitative analysis was then conducted for the 8.0 at.% Cu isoconcentration surface (referred to as the isosurface) inside the region of interest (ROI) shown in Figure 6h. The compositional profiles are displayed in a one-dimensional (1D) proximity histogram (referred to as a proxigram), as shown in Figure 6i. The proxigram measured the composition as a function of the distance from the isosurface of a given element, integrated over the interface at a 0.0-nm distance from a selected precipitate. Hence, the core precipitate composition was confirmed via observation on the rightmost side of the proxigram [34]. It was as follows: ~60 at.% Cu, ~10 at.% Ni, ~10 at.% Mn, and ~2 at.% Al.

The tendency of the precipitate to be enriched with Cu, Ni, Mn, and Al is related to the mixing enthalpy values of these elements in the Fe_71.25_(CoCrMnNi)_23.75_Cu_3_Al_2_ MEA system. The lower mixing enthalpy between Cu and Ni, Mn, and Al as compared to Co and Fe caused the precipitate to contain small amounts of these three elements, resulting in the formation of B2–Cu-Ni-Mn-Al co-precipitates [35]. For clarity, Figure 6j shows a magnified view of the 3D APT reconstruction of the Aged_7h sample. The Cu-rich regions, illustrated in orange, were concentrated in the centers of the precipitates and encapsulated with Ni (green), Mn (yellow), and Al (blue). Moreover, the Cu particles shown in Figure 6j exhibited a nearly fully spherical morphology with diameters on the X, Y, and Z axes of 5.79, 5.48, and 4.16 nm, respectively. This morphology indicates that the Cu-rich precipitate of this alloy was not affected by the coherency with the lath martensite structure in the alloy. In addition, B2-ordered Ni(Al, Mn) was found to play an essential role in stabilizing Cu-rich precipitates. Because the B2 Ni(Al, Mn) phase generally has a high melting point and low diffusivity, it may also act as a diffusion barrier to retard the growth of Cu-rich precipitates [30]. Consequently, nanosized Cu-rich precipitates formed, enhancing the hardness of the material. In addition, Figure 6k shows the 1D compositional profile of the different elements plotted along the cylinder axis shown in Figure 6j. In the present study, the addition of Cu and Al was intended to promote the formation of nanosized B2 intermetallic precipitates and Cu-rich precipitations during annealing, which could further improve the hardness, strength, and toughness of the alloy. The nanosized precipitates were observed in the aged samples and their size increased with increasing aging time. The formation of these precipitates is believed to be related to the diffusion of Cu and Al in the Fe-based matrix during annealing, which creates a thermodynamically favorable condition for the formation of the B2 and Cu-rich FCC phase. The presence of these precipitates could lead to a strengthening effect on the matrix, as they act as obstacles to the motion of dislocations, which in turn enhances the mechanical properties of the alloy. It is important to note that the current study solely focuses on the microstructural and mechanical properties of the Fe_71.25_(CoCrMnNi)_23.75_Cu_3_Al_2_ alloy after aging treatment. While our investigation offers valuable insights into the potential utilization of this alloy as a high-performance structural material, further investigation is needed to fully understand the underlying mechanisms of the phase transformations that occur during the aging process. Moreover, our study did not explore the effects of other variables such as the annealing temperature, cooling rate, or quenching media on the microstructural and mechanical properties of the alloy. Future studies could explore these variables to provide a more comprehensive understanding of the alloy’s behavior under different processing conditions. The underlying mechanisms of the formation of the nanosized precipitates and their impact on the mechanical properties of the alloy will be done and published in the near future.

## 4. Conclusions

Fe_71.25_(CoCrMnNi)_23.75_Cu_3_Al_2_ medium-entropy alloy was successfully developed through an aging treatment inspired by maraging steel. The alloy exhibited excellent mechanical properties with a peak Vickers microhardness of 597 HV, compressive yield strength of 2079 MPa, and ultimate compressive strength of 2843 MPa, achieved after aging at 450 °C for 7 h. The chemical composition of the alloy played a crucial role in determining the stability of the matrix phase, where Fe and Al acted as BCC stabilizers, and Cu promoted FCC phase formation. The microstructure of the aged alloy revealed the presence of lath martensite, consisting of BCC and B2 nanoprecipitates enriched with Cu, Ni, Mn, and Al. The formation of these nanosized precipitates with a diameter of approximately 5 nm resulted in a significant increase in the hardness and strength of the alloy. The findings of this study could provide insights into designing and developing novel high-strength and high-toughness alloys based on the principles of maraging steel, which can be used in various industrial applications.

## Figures and Tables

**Figure 1 materials-16-03589-f001:**
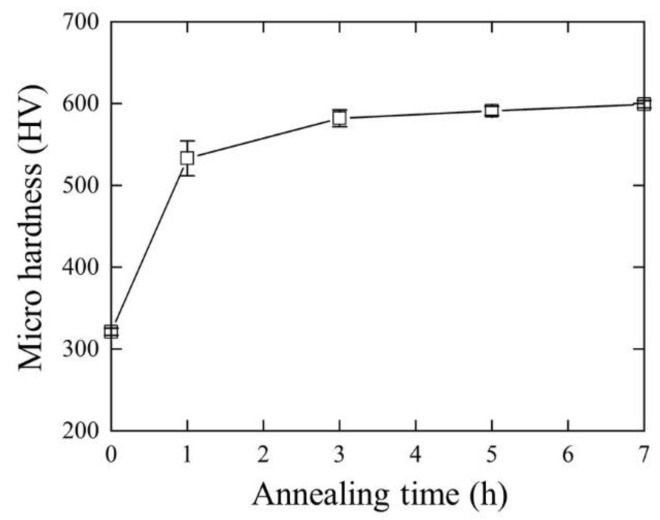
Micro-Vickers hardness of Fe_71.25_(CoCrMnNi)_23.75_Cu_3_Al_2_ (at.%) specimen at different annealing times.

**Figure 2 materials-16-03589-f002:**
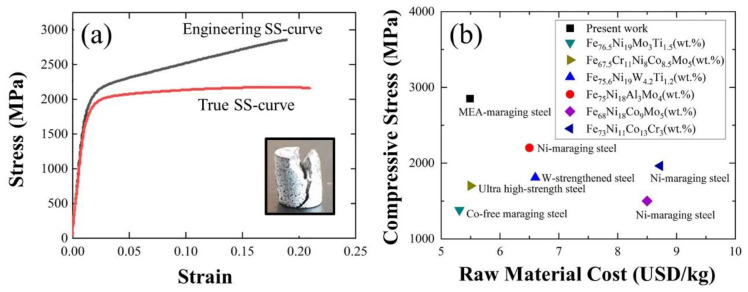
(**a**) Compressive properties of Aged_7h specimen. (**b**) Comparison of ultimate compressive strength (MPa) and raw material cost (USD/kg) of typical maraging steels and the novel alloy in the present work [5].

**Figure 3 materials-16-03589-f003:**
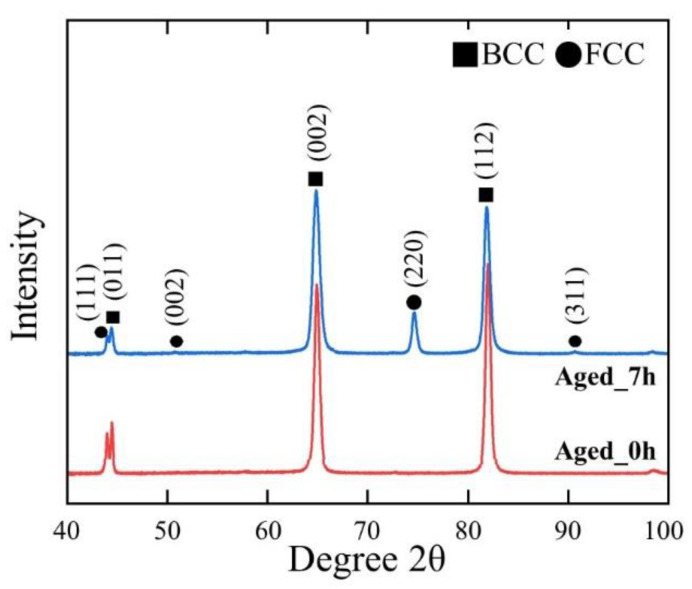
XRD patterns of Aged_0h and Aged_7h samples.

**Figure 4 materials-16-03589-f004:**
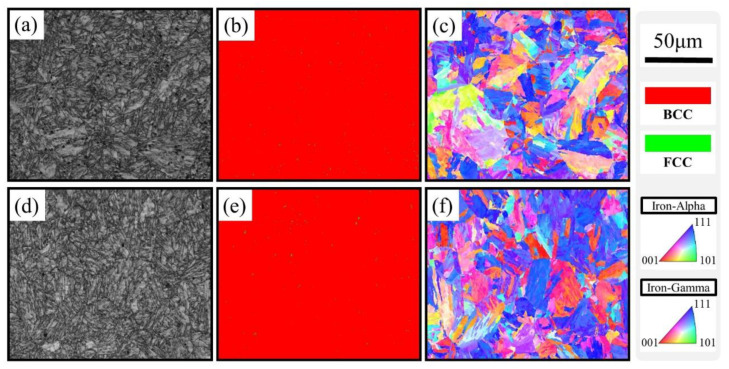
EBSD maps of Aged_0h (**a**–**c**) and Aged_7h (**d**–**f**) specimens. IQ maps (**a**,**d**); Phase maps (**b**,**e**); IPF maps (**c**,**f**).

**Figure 5 materials-16-03589-f005:**
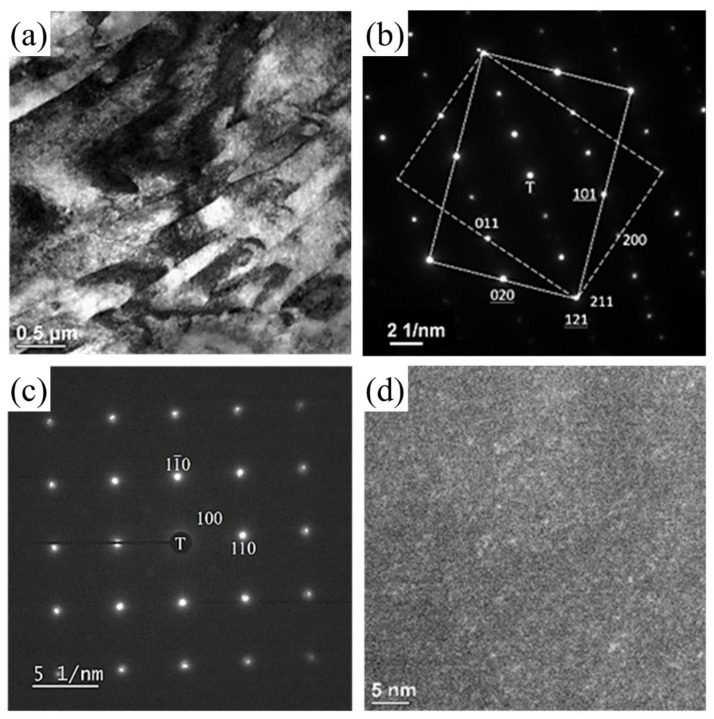
TEM micrographs of Aged_7h sample. (**a**) BF image showing ferrite lath grains; (**b**) SADP for two neighboring ferrite lath grains with a twinning relationship and the following zone axis directions: [$0\bar{1}1$]_t_//[$\bar{1}01$]_α_; (**c**) SADP along the [001] axis of the ferrite lath; (**d**) DF image from the reflection located near the (100) position in (**c**), showing the homogeneous distribution of the fine nanoprecipitates. T: transmission beam.

**Figure 6 materials-16-03589-f006:**
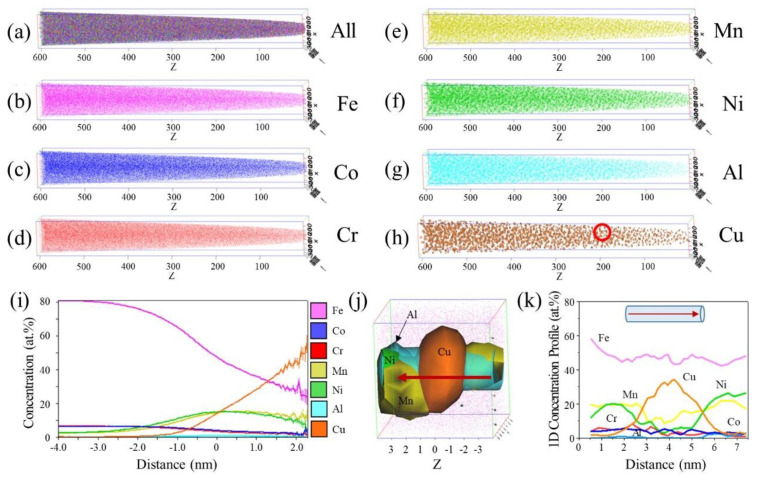
3D isosurface reconstruction of APT results for an analyzed Aged_7h sample volume of 77.6 nm × 74.3 nm × 607.4 nm. (**a**) All ions and (**b**–**h**) the ion map of each element; (**i**) 1D proxigram for 8 at.% Cu isosurface selection delineated in (**h**); (**j**) 3D reconstruction of APT results for the 8 nm × 8 nm × 8 nm cube dimensions corresponding to the Cu isosurface ROI delineated in (**h**), which is indicated by a red circle. The Cu isosurface (orange) was encapsulated by the Ni isosurface (green), Mn isosurface (yellow), and Al isosurface (blue); (**k**) 1D compositional profile for different elements, as plotted along the cylinder’s axis and corresponding to the red arrow in (**j**). 
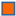
 Cu, 
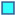
 Al, 
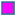
 Fe, 
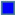
 Co, 
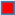
 Cr, 
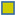
 Mn, and 
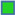
 Ni. (For an interpretation of references to the colors in this figure legend, the reader is referred to the web version of this article).

## Data Availability

Not applicable.

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
