# Peer review of "Microstructural Changes and Mechanical Properties of Precipitation-Strengthened Medium-Entropy Fe71.25(CoCrMnNi)23.75Cu3Al2 Maraging Alloy"

_materials, 2023, doi:10.3390/ma16093589_

Round 1

Reviewer 1 Report

In the Materials and Methods, the authors state that they used 1 kg F for the Vickers test. Why this specific force? It is rather low for testing the Vickers hardness of steel, or did they measure with a microhardness testing machine?
The results of the hardness measurements, Figure 1, are these average values in the graph? How many measurements did the authors make, and on which samples, i.e. where the samples were taken from the rolled material. Was the hardness measured via the surface or via the sample thickness? This is because they quenched the material, and are they sure that the quenching was done over the entire thickness of the material?

Also, in Materials and Methods, the authors state that "a hot compression test was performed at room temperature" - Could the authors elaborate a little on the test conditions? What temperature, compression speed, according to which standard?

Reviewer 2 Report

The study provides valuable insights into the microstructural changes and mechanical properties of precipitation-strengthened medium-entropy Fe71.25 (CoCrMnNi) 23.75Cu3Al2 maraging alloy. The results suggest that the formation of nanosized precipitates significantly enhances the hardness of the alloy, which can have important implications for various industrial applications. Nevertheless, some additions are needed:

1.    Abstract - provide some numerical values as properties. It looks general.

2.    The literature review is very descriptive and doesn't clearly highlight the current gaps in the field, which underpins the current investigation. More critical and comprehensive literature is needed.

3.    The criteria of investigated materials selection are not explained.

4.    Please clearly state the reasons for choosing the annealing times (1, 3, 5, 7) used in this study

5.    The results and discussion of the paper is good.

6.    The study could benefit from a more detailed discussion of the mechanisms underlying the formation of nanosized precipitates and their impact on the mechanical properties of the alloy. This would help to provide a more comprehensive understanding of the findings and their potential implications.

7.    The study could also benefit from a more detailed discussion of the limitations of the research approach and potential avenues for future research. This would help to provide a more nuanced perspective on the findings and their broader implications for the field.

8.    The conclusions are not concise. Please write the most important conclusions as brief points. Being as quantitative as possible. Do not just summarize what work was conducted in the manuscript.

9.    A few references need to be updated with some recent papers published in the last years.

10. The manuscript is overall well-written. However, there are many typesetting and grammatical errors in the text that should be corrected.

Reviewer 3 Report

The present study investigated the structure and mechanical property of Fe-Co-Cr-Mn-Ni-Cu-Al MEA during annealing. The authors give detailed analysis in microstructure, but several points should be addressed:

1.      The motivation of present study is not clear, why choose the present alloy, and this specific composition, what is the drawbacks of previous study on this material.

2.      More details on the determination of fabrication process is needed, why recrystallized at 900 degree, why annealing at 450 degree, why 1,3,5,7 hours

3.      Error bar is needed in fig.1

4.      Mechanical property only includes Aged-7h, what about other samples. The author should give comprehensive analysis on all samples.

5.      Fig.2(b) give the cost of alloys, how to calculate, is it correct, please give more details

6.      In fig.4, FCC is hardly to find, but in XRD, the peak from FCC is obvious, why

7.      The analysis on the microstructure is in chaos, difficult to understand, where is the nanoparticle, what is the volume fraction of each phase (BCC,FCC,nano-particle), when fcc is formed, and when nanoparticle is formed, how to affect the BCC matrix, finally, how does these complex phase formation affect the mechanical property.

Based on above points, I suggest MAJOR REVISION on this paper.

Round 2

Reviewer 2 Report

Accept

Reviewer 3 Report

The points have been well responded